# Macroscopic helical chirality and self-motion of hierarchical self-assemblies induced by enantiomeric small molecules

Yang Yang[1,2], Jie Liang[1], Fei Pan[1,3], Zhen Wang[1,2], Jianqi Zhang[1], Kamran Amin[1,2], Jin Fang [1], Wenjun Zou[1], Yuli Chen [3], Xinghua Shi [1,2] & Zhixiang Wei[1,2]

Transfer of molecular chirality to supramolecular chirality at nanoscale and microscale by chemical self-assembly has been studied intensively for years. However, how such molecular chirality further transfers to the macroscale along the same path remains elusive. Here we reveal how the chirality from molecular level transfers to macroscopic level via self-assembly. We assemble a macrostripe using enantiomeric camphorsulfonic acid (CSA)-doped polyaniline with hierarchical order. The stripe can twist into a single-handed helical ribbon via helical self-motion. A multi-scale chemo-mechanical model is used to elucidate the mechanism underlying its chirality transfer and induction. The molecular origin of this macroscopic helical chirality is verified. Results provide a comprehensive understanding of hierarchical chirality transfer and helical motion in self-assembled materials and even their natural analogues. The stripe exhibits disparate actuation behaviour under stimuli of enantiomeric amines and integrating such chiral perception with helical self-motion may motivate chiral biomimetic studies of smart materials.

[1] CAS Key Laboratory of Nanosystem and Hierarchical Fabrication, CAS Center for Excellence in Nanoscience, National Center for Nanoscience and Technology, 100190 Beijing, China. [2] University of Chinese Academy of Sciences, 100049 Beijing, China. [3] Institute of Solid Mechanics, Beihang University, 100191 Beijing, China. These authors contributed equally: Yang Yang, Jie Liang, Fei Pan. Correspondence and requests for materials should be addressed to X.S. (email: shixh@nanoctr.cn) or to Z.W. (email: weizx@nanoctr.cn)

Symmetry breaking is responsible for many physical, chemical, and biological phenomena[1]. Mirror symmetry breaking leads to chirality, one of the key structural features of natural systems at different scales[2]. Although the origin and induction of chirality has not been fully clarified, some specific genes and their expression processes are thought to determine handedness in biological systems[3,4]. Inspired by intriguing helical assemblies in nature, scientists have endeavored in creating helical architectures of varying lengths via top-down or bottom-up approaches[5]. Among various approaches, self-assembly is one of the most powerful techniques to construct order structures beyond molecular level[6–9]. Evidently, self-assembly is suitable for fabricating chiral materials as a result of progressive chirality induction and transfer during ordered assembly[10–13]. However, the helicity of most of these self-assembled structures is normally limited in the nano and microscale due to their order limitation. Usually state-of-the-art macroscopic helical structures can be well constructed via top-down approaches using elastomers[14,15], liquid crystals[16–18], or hydrogels[19,20]. Accordingly, leaving the transfer of molecular chirality further to the macroscopic chirality via self-assembly a great scientific challenge, creating an obstacle to deep understanding of the corresponding chirality transfer mechanism as well[5].

Helical motion is dedicated to numerous significant processes in living organisms[21,22]. Once a material with controllable helical sense is constructed, its potential to achieve helical motion mimicking its natural analogues may fully be accomplished. Stimuli-response is widely existing in living organisms, swelling/shrinking induced shape change is a typical example, in which subtle intermolecular or inter-substructure motions induce transformation in their macroscopic textures[23]. Hierarchical order in these organisms is beneficial to enhance their responsive sensitivity and accuracy, which is inspiring material and device fabrication with biomimetic advantages[24–29]. Similarly, chirality perception in natural supramolecular systems such as enzymes, proteins, and DNA arousing biomimetic studies on artificial supramolecular chiral sensing[30]. However, achieving macroscopic helical self-motion in stimuli-responsive material is still a challenge[31]. Thus integrating chirality perception with helical self-motion is likely to develop a kind of stimuli-responsive system.

In this paper, using conjugated polyaniline (PANI) as a representative, we report how molecular chirality of its enantiomeric dopants, breaking through the scale barriers, transfers to macroscopic helical chirality of PANI ribbons. Chirality induction in polyaniline by enantiomeric camphor sulfonic acid (*r*-CSA and *s*-CSA) as a dopant has been studied meticulously in its molecular[32], nanoscales, and microscales[33], nevertheless, macroscopic PANI assembly with controllable helicity is never investigated so far (Fig. 1a). The process for material preparation and its macroscopic helicity generation is illustrating in the Fig. 1b. In the first step, we doped PANI with enantiomeric CSA into single-handed polymer (PANI:*r*-CSA or PANI:*s*-CSA) during polymerization of aniline monomer. Subsequently, the PANI:CSA was assembled into helical fibrous microscopic assemblies, which further assembled into a macromembrane on a uniaxial stretched polypropylene substrate. Then the membrane was peeled down in tetrahydrofuran (THF) into flat macrostripes. In the second step, isopropanol (iPrOH) was then introduced in THF, curling the macrostripe into a single-handed helical ribbon, where its helical chirality at macroscale emerged. This molecular chirality transfers to the macroscopic chirality by triggering chiral impetuses in both the molecular and microscopic scale. A chiral stimuli-responsive system rises in response to the helical self-motion behaviour of the stripe under stimuli of chiral target species.

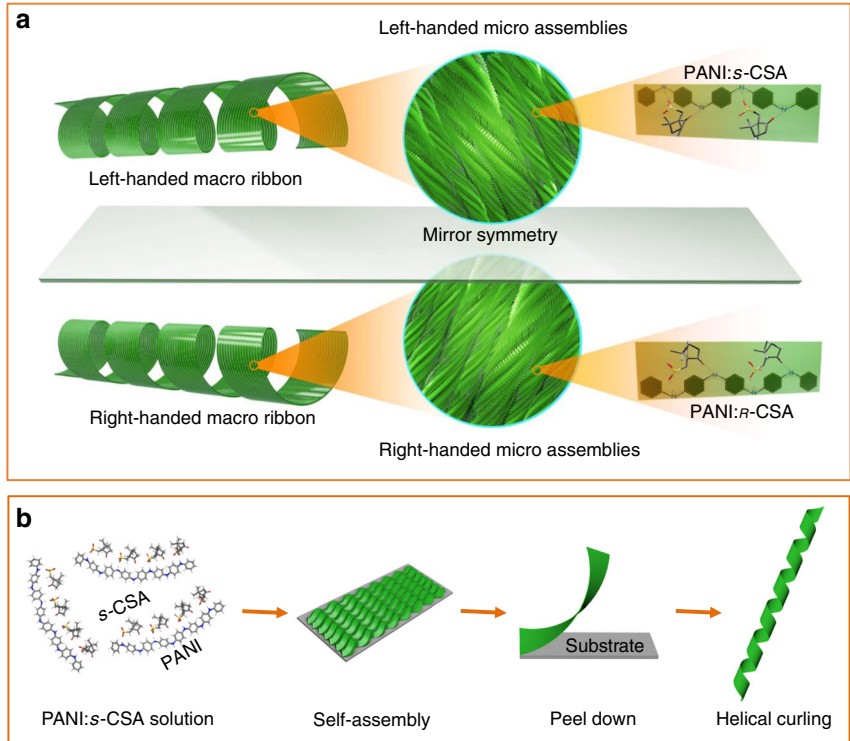

**Fig. 1** Illustration for the structure and preparation of the macroscopic assemblies. **a** Hierarchical components of the macroribbon. The mirror symmetrical left- and right-handed macroribbons (left) consisted of aligned left-handed and right-handed nano-assemblies and micro-assemblies (middle) and single-handed PANI polymer doped by *s*-CSA and *r*-CSA (right), respectively. **b** In-situ doping PANI with enantiomeric CSA-dopant by polymerization of aniline monomer in organic solvent; self-assembly of PANI molecules and its microscopic fibrous assemblies on the uniaxial stretched PP substrate; peel down the PANI macrostripe from the PP substrate in THF solvent; curl the macrostripe into helical ribbon with adding iPrOH

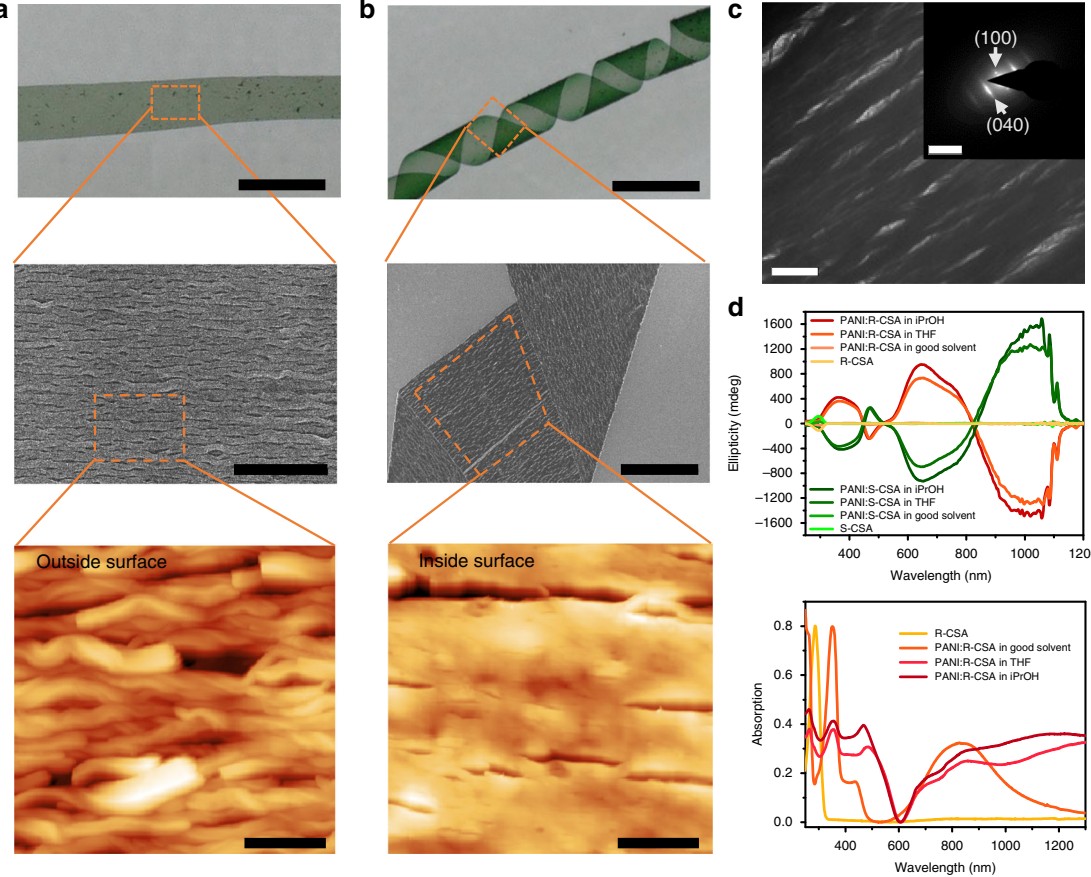

**Fig. 2** Structural characteristics and basic properties of the PANI macroribbon. **a** Photograph of the macro stripe peeled down in THF. **b** Photograph of macro helical ribbon that resulted from the stripe after adding iPrOH. Scale bars corresponds 0.5 mm. SEM and AFM images show the aligned fibrous nano and micro assemblies in the stripe. As shown in the AFM images, assemblies on the outside surface of the stripe were single-handed helical (**a**) but straight on the inside surface (**b**). Scale bars of photographs in **a** and **b** correspond 0.5 mm; scale bar of SEM image in **a** corresponds 10 μm, and scale bar of SEM image in **b** corresponds 50 μm, respectively; scale bar of AFM images in both **a** and **b** correspond 3 μm. **c** TEM image and corresponding SAED pattern of the stripe. Scale bar in TEM image corresponds 2 μm and in SAED corresponds 5 nm⁻¹. **d** CD and UV–vis–NIR spectra of the CSA and PANI:CSA dissolved in methanol or CHCl₃/THF; the helical assemblies of the PANI ribbons dispersed in THF and iPrOH, respectively

## Results

**Basic characters of the macrostripe.** In detail, since the stripe was assembled on the polypropylene substrate, the microscopic morphologies of outside surface and inside surface of the stripe are different. Outside surface is the surface toward the solution during self-assembly, whereas inside surface is the counter-surface adhering to the polypropylene substrate. The fibrous microscopic assemblies of the outside surface are uniform single-handed helices, whereas those of the inside surface are almost straight due to the restriction of the polypropylene substrate (scanning electron microscopy and atomic force microscopy in Fig. 2a, b). The alignment between these microscopic fibrous assemblies originated from the molecular and its crystallographic orientation of the polypropylene substrate (Supplementary Fig. 1). Meanwhile, face to face π–π stacking between PANI molecules has given rise to the intermolecular order in those fibrous assemblies[34]. This long-range intermolecular order in the stripe is proven by the sharp diffraction points in the selected area electron diffraction (SAED; Fig. 2c, and details in Supplementary Note 1). This result, combined with its corresponding transmission electron microscopy (TEM) image, revealed that the direction of the intermolecular π-π stacking was along the long axis of the fibrous assemblies as well as the stripe. The arrangements of both the molecules and fibrous assemblies were further proven by

the aligned diffraction fringe in wide angle X-ray scattering (WAXS) of the stripe (Supplementary Fig. 2).

The circular dichroism (CD) spectrum and its corresponding ultraviolet–visible–near-infrared (UV–vis–NIR) spectrum were employed to study the helicity of the macrostripe (Fig. 2d). The enantiomeric CSA shows the CD band at ca. 295 nm. Before self-assembly, PANI:CSA solution shows three weak CD bands at ca. 400, 460, and 600 nm (enlarged image in Supplementary Fig. 3). Whereas, the macrostripe possesses the CD bands at *ca.* 360 and 470 nm, and its bisignate Cotton bands at *ca.* 650 and 1000 nm corresponded to the absorption at ca. 830 nm in the UV-vis-NIR spectrum. The CD spectra of *R*-CSA and *s*-CSA, as well as the corresponding supramolecular assemblies, were symmetrical, which corresponded to the symmetric helical directions of the macroribbons. Note that the bisignate CD bands at 650 and 1000 nm, which were attributed to the supramolecular chirality from the chiral exciton coupling of the interchain helical stacking of PANI molecules in the stripe[32], were distinctly enhanced when the stripe was emerged in iPrOH. Due to their structure or conformation difference[32,35–37], the chiroptic features of PANI:CSA and its assemblies here are different from PANI:CSA aggregates prepared in aqueous-phase electrochemically[38,39] or chemically[40]. The difference in molecular structure and conformation of PANI:CSA further results in their different

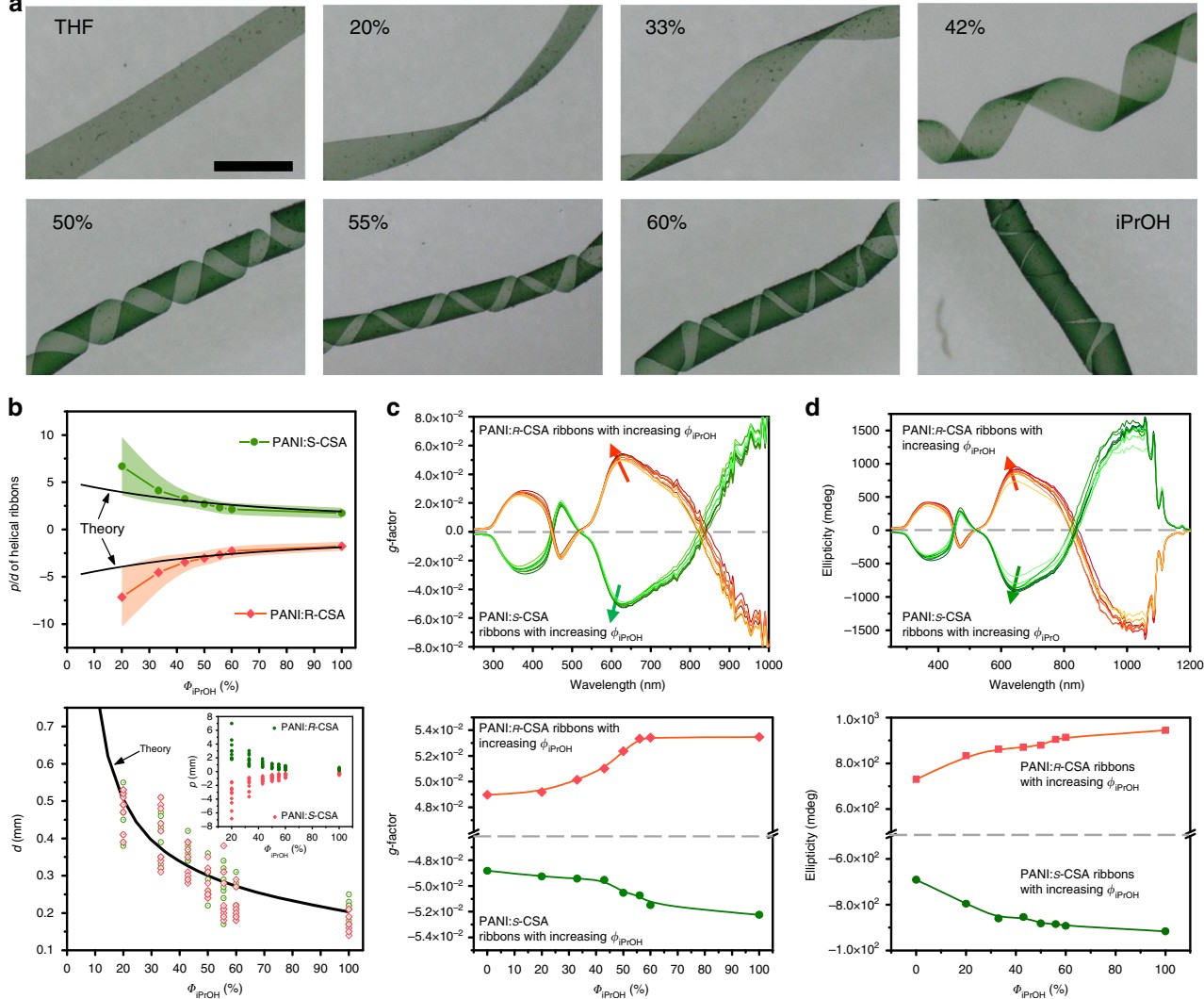

**Fig. 3** Macroscopic helical chirality generation in the stripe. **a** Photographs of the left-handed helical ribbons with different feature size of helicity under different $\Phi_{iPrOH}$. By introducing iPrOH into THF, the PANI:$s$-CSA macro stripe shrunk into left-handed helical ribbon step-by-step. Scale bar corresponds 0.5 mm. **b** Feature size of helicity $p$ (helical pitch), $d$ (diameter of the cylindrical helix), and proportional relation $p/d$ of the helical ribbons decreased with increasing $\Phi_{iPrOH}$. The black lines are theoretical calculating values of the feature size of helicity. **c** $g$-factor, and **d** the corresponding CD of the PANI:$s$-CSA and PANI:$R$-CSA ribbons measured in the co-solvent with different $\Phi_{iPrOH}$

intermolecular stacking, which aggravated the difference in their characteristic CD bands (more details see Supplementary Note 2).

**Helical chirality generating in the macrostripe**. In the second step, the helical chirality induction of the macrostripe occurred in a controllable manner by introducing iPrOH into THF proportionally (Fig. 3a). Its feature size of helicity $p$ and $d$ (helical pitch $p$ and diameter $d$ of the cylindrical helix) and proportional relation $p/d$ decreases with increasing $\Phi_{iPrOH}$ (the ratio of iPrOH to the total volume of iPrOH and THF; Fig. 3b). Before adding iPrOH, the PANI:$s$-CSA stripe was almost flat in THF. With $\Phi_{iPrOH}$ increased to 20%, the stripe began to curl slightly along a left-handed helical direction. Further increasing $\Phi_{iPrOH}$ from 33 to 55% induced an explicit left-handed helical ribbon with decreasing feature size of helicity, eventually forming a tubular helix with 60% or higher $\Phi_{iPrOH}$. Upon re-immersing into THF, the helical ribbon turned to its flat shape and vice versa (Supplementary Movie 1). A similar situation was observed in the PANI:$R$-CSA macrostripe, but a right-handed helical ribbon was instead generated (Supplementary Fig. 4). We introduced $g$-factor

to quantitatively characterise the helicity of this chiral system along with its corresponding CD[41], because $g$-factor is independent of sample concentration and optical path length[42]. The $g$-factor and corresponding CD of the PANI:$s$-CSA and PANI:$R$-CSA ribbons emerged in the co-solvent with different $\Phi_{iPrOH}$ were measured (Fig. 3c, d). As $\Phi_{iPrOH}$ increased, the $g$-factor at around 650 nm increased along a negative direction reached from $-4.882 \times 10^{-2}$ to $-5.224 \times 10^{-2}$ in PANI:$s$-CSA ribbons. Same parameter that persistently increased along the opposite direction reached from $4.897 \times 10^{-2}$ to $5.347 \times 10^{-2}$ in PANI:$R$-CSA ribbons. Given that iPrOH is a poorer solvent than THF for PANI, introducing iPrOH into THF induced an intermolecular shrinkage in the stripe (Supplementary Fig. 7 and DFT calculation see Supplementary Note 3), which in turn enhanced the CD intensity and corresponding $g$-factor attributed to the intermolecular chiral exciton coupling enhancement[43].

**Chirality induction mechanism based the multi-scale chemomechanical model**. The mechanism underlying chirality induction in the helical ribbon is elucidated by the multi-scale

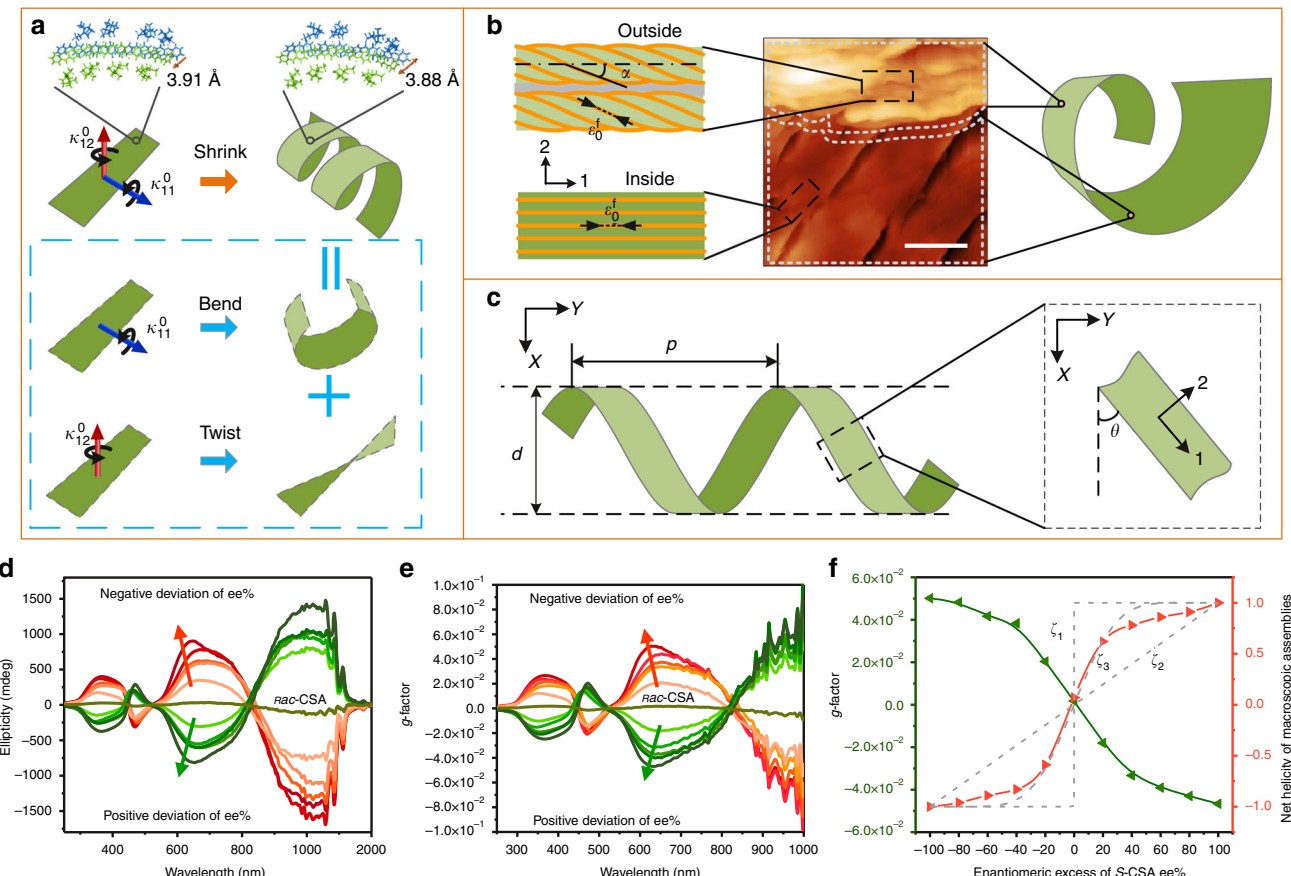

**Fig. 4** Amplification mechanism and molecular origin of the macroscopic helical chirality in the macro ribbons. **a** Intermolecular shrinkage (**a**) induced the microscopic shrinkage strain $\varepsilon_0^f$ along the fibrous nano assemblies in **b**. **b** The directions of $\varepsilon_0^f$ related to the helical directions of the microscopic fibrous assemblies in the stripe. The direction difference of $\varepsilon_0^f$ between the outside and inside surfaces inducing normal and shear strain gradients simultaneously along the thickness that in turn induces the spontaneous bending curvature $\kappa_{11}^0$ and torsional curvature $\kappa_{12}^0$, respectively. Also, the curvatures $\kappa_{11}^0$ and $\kappa_{12}^0$ are related to the helical angle $\alpha$. The curvature decomposition in dashed blue box in **a** illustrate that $\kappa_{11}^0$ induces the stripe bending, whereas $\kappa_{12}^0$ induces its twisting. **c** Guiding by the curvatures $\kappa_{11}^0$ and $\kappa_{12}^0$, the stripe curls into a helical ribbon with its feature size of helicity $p$ and $d$. The bending curvature $\kappa_{11}^0$ induces the inside surface to bend toward the inner side of the helical ribbon. Scale bar of AFM in **b** corresponds 3 μm. **d** CD spectra, and **e** g-factor spectra of a series of stripes doped by certain enantiomeric excess of s-CSA. **f** Majority rule experiments: the disproportionate relationship between g-factor at ca. 650 nm of the ribbons and enantiomeric excesses of s-CSA (green line), and the disproportionate relationship between net helicity of the ribbons and enantiomeric excesses of s-CSA (aurantia line)

chemo-mechanical theoretical model (Fig. 4). At the molecular level, introducing iPrOH into THF results in the intermolecular shrinkage in the stripe (Fig. 4a). The intermolecular shrinkage perpendicular to the π-plane of PANI induces the microscopic shrinkage strain $\varepsilon_0^f$ along the long axis of the fibrous nano-assemblies in the stripe. However, the directions of $\varepsilon_0^f$ on the two surfaces also differ due to the variation in microscopic fibrous morphology between the outside and inside surfaces, resulting in a macroscopic strain difference between the two opposite surfaces (Fig. 4b). Particularly, the difference in macroscopic strain simultaneously induces normal and shear strain gradients along the thickness, which led to macroscopic spontaneous bending curvature $\kappa_{11}^0$ and torsional curvature $\kappa_{12}^0$, respectively (Fig. 4a). As these gradients are related to the helical angle $\alpha$ between the nano-assemblies and the direction 1 in the outside surface, the thickness of ribbon $h$, and $\Phi_{iPrOH}$, the two curvatures can be expressed as

$$\kappa_{11}^0 = -\frac{\varepsilon_0^f \sin^2 \alpha}{h} \quad \text{and} \quad \kappa_{12}^0 = \frac{0.4}{\Phi_{iPrOH} + 0.4} \frac{\varepsilon_0^f \sin 2\alpha}{h} \quad (1)$$

Guided by the two curvatures, the stripe curled into a cylindrical helix with specific feature size of helicity $p$ and $d$ (top and side view in Fig. 4b, c; theoretical calculation results in Fig. 3b; and detailed in Supplementary Note 4). The bending curvature $\kappa_{11}^0$ induces the flat inside surface to bend toward the inner side of the macrohelix, whereas the torsional curvature $\kappa_{12}^0$ induces the stripe twisting (curvature decomposed in the dashed blue box in Fig. 4a). The helical direction (i.e., the chirality) of the ribbon was determined by the sign of the torsional curvature $\kappa_{12}^0$, namely, the handedness of the fibrous microscopic assemblies in the ribbon. Hierarchical order in the stripe ensured the intermolecular shrinkage induced strain transfer from bottom up to the macroscopic level.

From the view of energy, the out-of-plane deformation energy density of the macroribbon can be expressed as[44]

$$E_{out} = \frac{1}{2} \mathbf{K} \mathbf{D} \mathbf{K}^T, \quad (2)$$

where $\mathbf{K} = \left[ \kappa_{11} - \kappa_{11}^0, \kappa_{22} - \kappa_{22}^0, \kappa_{12} - \kappa_{12}^0 \right]$ is the effective bending/torsional curvature that can induce elastic strain energy, and

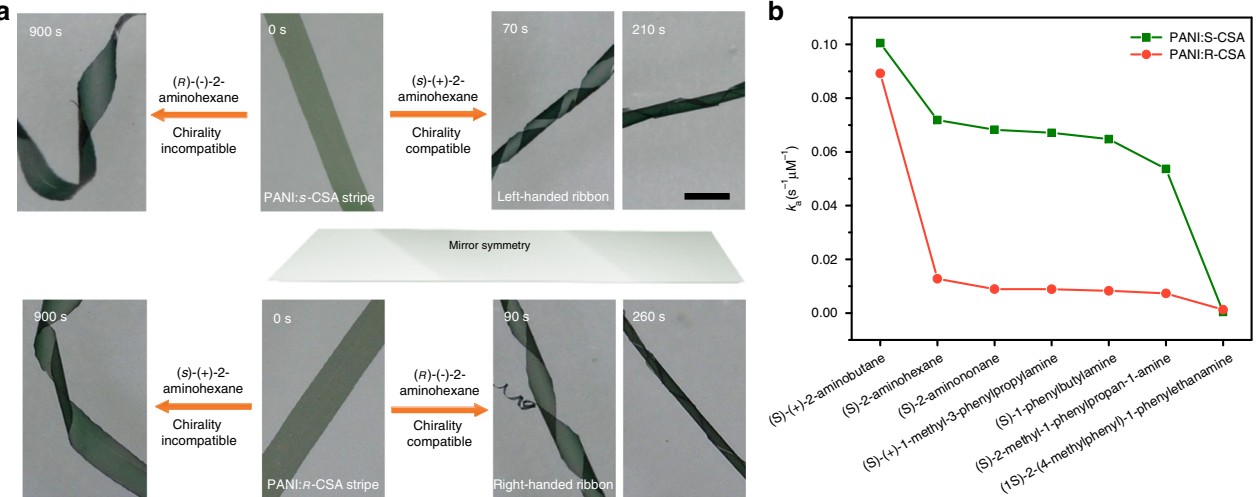

**Fig. 5** Chirality perception of the macro stripe. **a** The upper half, the PANI: s-CSA stripe exposed to (s)-(+)-2-aminohexane and (R)-(-)-2-aminohexane in THF. The bottom half, the PANI:R-CSA stripe exposed to (s)-(+)-2-aminohexane and (R)-(-)-2-aminohexane in THF. Disparate actuation behaviours over time can be observed when the stripes responded to chiral stimuli from different enantiomeric aminohexanes. Scale bar corresponds 0.5 mm. **b** the de-doping apparent rate constant $k_a$ of a series of chiral target amine species for PANI stripes doped by enantiomeric CSA

$D$ is the out-of-plane stiffness matrix that is related to the thickness of the ribbon $h$ and the orthotropic elastic constants matrix of the material. Under a certain condition, the energy is a function of the helical pitch $p$ and diameter $d$ of the macrohelix, as shown in the following expression:

$$E_{\mathrm{out}} = f(p, d) \qquad (3)$$

Accordingly, with increasing $\Phi_{\mathrm{iPrOH}}$, the large energy of the flat stripe ($d\rightarrow\infty$) leads to an unstable state in it; thus, the flat stripe curls into a helical ribbon with certain $p$ and $d$ to reach a relatively stable status and attain the energy $E_{\mathrm{out}}$ minimum.

**Molecular origin of the macro helical chirality**. According to the above analysis, the helical directions of both the macroribbons and their microscopic assemblies correspond to the chirality of the enantiomeric CSA-dopants. Therefore, majority rules experiments were performed to probe whether chirality amplification exhibits in the present system[45,46]. A series of helical ribbons doped by certain enantiomeric excess of s-CSA (([s-CSA]-[R-CSA])/([s-CSA] + [R-CSA]), ee%) were prepared, and CD and g-factor of these ribbons were measured as well as their microscopic morphology was recorded by SEM (Fig. 4d, e, and Supplementary Fig. 10). When the enantiomeric excess of s-CSA deviated either positively or negatively from 0, the g-factor and CD intensity of the ribbons increased in opposite directions, nonlinearly. The sigmoid nonlinear relationship between g-factor intensity at ca. 650 nm and the enantiomeric excess of s-CSA (green line in Fig. 4f), for example, reveals the existence of a typical chirality amplification phenomenon. The nonlinear effect suggests that the majority of enantiomeric CSA dictates the helicity of the supramolecular assemblies[41,47]. The chirality amplification occurred in the course of supramolecular self-assembly in the majority rules experiments, while the disproportionate change in the net helicity of both macro and micro assemblies caused by ee% of s-CSA were kept, in virtue of chirality induction in the subsequent steps (the aurantia line in Fig. 4f, Supplementary Fig. 11a, detail see Supplementary Note 5). Evidently, the molecular chirality of enantiomeric CSA transfers to the supramolecular chirality of the PANI:CSA assemblies via

the principle of supramolecular self-assembly, which determines the helical direction of the microscopic assemblies[32,48]. The helicity of the microscopic assemblies further induced the helical chirality of the macroscopic ribbons while its shrink. The macroscopic helical chirality of the ribbons stems from the molecular chirality of the enantiomeric CSA who commands the helicity of the entire system, progressively.

**Helical self-motion of the macrostripe under stimuli of enantiomeric aminohexanes**. Discrimination of enantiomers, such as amines[49,50], diamines[51], alcohols[52,53], amino alcohols[54,55], amino acids[56], and the like that closely related to pharmaceutical, chemical, biotechnology and bio-safety, environment, and food industries, is one of core research fields in analytical and catalytic chemistry[57-60]. With hierarchical chirality, the macrostripes demonstrated unique helical self-motion behaviours when enantiomeric amines are employed as target species to study their enantioselective discrimination ability in THF solvent (Fig. 5). When PANI:s-CSA stripe exposed to (s)-(+)-2-aminohexane, for example, the stripe immediately turned from green to blue while exhibiting a left-handed helical motion. By contrast, when PANI:s-CSA stripe exposed to (R)-(−)-2-aminohexane, the stripe slowly changed color, and an unpredictable deformation is observed. Similarly, the PANI:R-CSA stripe turned to blue but right-handed helical motion was observed when exposed to (R)-(−)-2-aminohexane (Fig. 5a).

In the THF solvent, chirality compatible target enantiomers can readily migrate into the helically stacked PANI:CSA assemblies. After which, the target enantiomer anchors itself with the neighboring CSA smoothly through the amino and sulfonic groups[61]. Whereas chirality incompatible target enantiomers can neither be migrated into the helically stacked PANI:CSA assemblies easily nor anchors itself with neighboring CSA. This makes the de-doping apparent rate constant $k_a$ of chirality compatible target enantiomers is an order of magnitude higher than that of chirality incompatible target enantiomers (Fig. 5b, details see Supplementary Note 6). High $k_a$ of chirality compatible target enantiomers leads to complete de-doping of CSA from the PANI backbone in advance of distinct deformation of the stripe, which results in quickly color change as well as chiral collapse in the PANI:CSA assemblies. As a result, the helical motion

occurred in the stripe while uniform shrinking. However, low $k_a$ of chirality incompatible target enantiomers can but lead to incomplete and uneven de-doping of CSA from the PANI backbone in advance of the distinct deformation of the stripe. This gives rise to disordered deformation of the stripe (Supplementary Movie 2). Regarding the amino as the pivot of the target species, besides its configuration, the substituent groups attached to its two sides influence its $k_a$ by influencing both its migration into PANI:CSA assemblies and its bonding to enantiomeric CSA due to the steric hindrance effect. As the volume of the substituent increases, $k_a$ decreased significantly (Fig. 5b). With too low or too high steric hindrance from the substituent groups, the enantiomeric amines will lose their selectivity to the stripe. On the other hand, the deformation behaviour of the stripe is sensitive to the concentration of the target species, low initial concentration of amino target species only leads to mild irregular deformation of the stripes.

## Discussion

We fabricated a macrostripe with hierarchical order via a chemical self-assembly strategy and molecular chirality of the enantiomeric CSA dopant induced macroscopic helical chirality of the stripe. We proved that the macroscopic helical chirality can merely root from the molecular chirality in ordered self-assembled material. Such molecular chirality is likely to serve widely as chirality origin in self-assembly of chiral structures at different scales. Particularly, we demonstrated how this molecular chirality can be transferred to the macroscopic chirality by triggering chiral impetuses in both the molecular scale and microscopic scale, which may be used to understand motion behaviours of a main category of artificial and natural helical structures, analogous to the system used in this study. The helical self-motion behaviour of the stripe under stimuli, by both solution and chiral species, may provide design standpoints for smart materials with accurate molecular or substructure organizations to achieve complex motions.

## Methods

**Synthesis of single-handed PANI molecules**. Synthesis of single-handed PANI molecules is similar to our previous reports with slight modification[34,61]. Aniline was purified by reduced pressure distillation, and THF was distilled with sodium before using. 2,3-Dichloro-5,6-dicyano-1,4-benzoquione (DDQ), chloroform, and s-CSA or R-CSA were used as received. Firstly, s-CSA or R-CSA (1.6 molar ratio of aniline) was dissolved in 5.0 mL of chloroform, and aniline (56.9 μL, 0.625 mmol) and oligomer N-phenyl-p-phenylenediamine (1/100 molar ratio of aniline) were added to the solution. DDQ (142 mg, 0.625 mmol) was pre-dissolved in 1.675 mL of THF. These solutions were vigorously shaken for full dissolution respectively, and then left undistured for 1 h. Secondly, the above-mentioned two solutions were mixed and shaken quickly for homogeneous mixing and kept at 22 °C for 6 h undistured for polymerization. As-prepared solution of helical polyaniline was diluted in good solvent (mixture of THF/CHCl$_3$ with a volume ratio of 1/3) for CD (Jasco J1500) and UV–vis–IR (Shimadzu, UV-3600) measurements.

**Self-assembly of PANI macro ribbons**. As-prepared chiral polyaniline (0.1 mL) was first diluted in 1.8 mL of good solvent in a test tube in the presence of a uniaxial stretched polypropylene (PP) template. Then, 2.2 mL of poor solvent methanol was added into the above solution. The mixture was vigorously shaken for 1 min and left to stand for days for self-assembly. As-prepared PANI membrane adhere to the PP template was peeled down in THF into the flat macro stripe, and the stripe transformed into a helical macroribbon in THF/iPrOH co-solvents. These macrostructures were recorded by optical microscopy (VHX-5000, Keyence). The stripe or ribbon was transferred on silicon wafer or carbon-coated copper grid for SEM (S-4800 or S-8220, Hitachi, Japan), AFM (M-Pico, Multimode Veeco), TEM and SAED (Tecnai G2 F20U-TWIN, FEI Co., USA) characterization. Dried samples transferred on copper grid and stripes immersed in the solvents of THF and iPrOH in quartz capillaries were used for WAXS characterization, respectively (Xeuss small-angle and wide-angle X-ray scattering system with Xenocs Cu Kα X-ray source GeniX Cu ULD and Dectris 100 K Pilatus). The ribbon dispersed in co-solvent with different $\Phi_{iPrOH}$ were characterised by CD and UV-vis-IR spectra.

**Majority rule experiments**. A series of PANI molecules was synthesised with certain enantiomeric excess of s-CSA and further self-assembled into macrostripes by the method mentioned above. These stripes curled into single-handed helical ribbons after submerging into mixed solvent with $\Phi_{iPrOH}$ fixed at 50%, and the net helicity of the helical ribbons was counted for each sample. And the CD and corresponding g-factor were measured for each sample. The results and statistics are illustrated in Fig. 4d–f.

**Actuation behaviour of the macro stripe exposed to chiral species**. About 1 μL of enantiomeric amine was pre-mixed in 200 μL of THF. Enantioselective actuation behaviours were observed after injecting the above solution into 4 mL of THF rapidly (the initial concentration of the enantiomeric amine is about 0.05 μM in total THF volume), in which the flat stripe curled along a single handedness or irregularly. The actuation behaviours of the macrostripes were recorded by optical microscopy.

A series of enantiomeric amines with different chemical structures and different initial concentrations were added into the suspensions of PANI:CSA stripes rapidly while performing the UV–vis–NIR spectra measurement. And the absorption of the PANI:CSA stripes at wavelength of 300–600 nm was recorded at set intervals. The attenuation of peak intensity at 470 nm of the PANI:CSA stripe with a given volume was used to demarcate its de-doping degree.

## Data availability

The data that support the finding of this study are available from the corresponding author on reasonable request.

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

## Acknowledgements

The authors acknowledge the financial support from the Ministry of Science and Technology of China (Grant No.2016YFA0200700), the National Natural Science Foundation of China (Grant Nos. 21534003, 91427302, 21603044, 11422215, and 11672079), and the Chinese Academy of Sciences.

## Author Contributions

Z.X.W. planned the project and supervised the research. Y.Y. and J.L. designed and conducted the experiments and data analysis, as well as proposed and developed the mechanism hypothesis together with Z.X.W.. F.P. and Y.Y. developed the mechanical model and calculations under the supervision of X.H.S.. Z.W. conducted the DFT calculation. J.Q.Z. conducted X-ray Scattering characterization. J.F. conducted some property characterization. W.J.Z. conducted the chirality perception experiments. Y.Y. and Z.X.W. wrote the manuscript, with contributions from X.H.S., F.P., K.A., and Y.L.C.. Y.Y., J.L., and F.P. contributed equally to this work.

## Additional information

**Competing interests:** The authors declare no competing interests.

