## [Peer Review File · Nature Communications]

Reviewers' comments:

Reviewer #1 (Remarks to the Author):

The authors prepared a kind of macro helical ribbon based on enantiomer-doped polyaniline, and discussed this chirality amplification phenomenon from molecular level to macroscopic level. In addition, they also found the helical self-motion of the ribbons under stimuli of chiral molecules. In general, it is a well-organized paper with a good description. As for an extensively-studied chiral conducting system, I don't think the authors have achieved some breakthroughs here. The detailed comments are listed below:

1. The authors have discussed the mechanism for the formation of helical ribbons from the view of energy, and clarified the relationship between some important parameters, including microscopic shrinkage strain, bending curvature, torsional curvature, and out-of-plane deformation energy. However, the above discussion part just explained why flat stripes will twist in poor solvent. The majority rule experiment and CD data are far from revealing the chirality amplification mechanism. Is it possible to find the functional relationship between the molecular-chirality and macro-helicity? The authors should strengthen this part.
2. The authors mentioned that the difference between the outside and inside surfaces contributed to the helical conversion. Will the choice of substrates for assembly influences the structure or p/d of the macro ribbon?
3. The helical self-motion is very interesting and inspiring. In my opinion, compared with the structure analysis, the should focus on the chirality-responsive property of these materials. For example, the authors can investigate how the structure of chiral molecules, concentration, and reaction time influence the helical conversion of ribbons from the view of thermodynamics and kinetics, which will motivate the study of biomimetic smart materials.
4. The authors should provide the scale bars for Figure 1 more clearly.

Reviewer #2 (Remarks to the Author):

The manuscript by Yang et al. describes the hierarchical self-assembly of PANI stripes prepared upon the addition of chiral CSA dopants. The presence of the chiral dopant induces the formation of chiral structures (demonstrated by the corresponding CD spectra). This phenomenon is well-reported in literature, including some references from the same authors not mentioned in the manuscript. The deposition of such PANI-CSA polymers onto a polypropylene substrate allows the preparation of stripes with dissimilar inside and outside surfaces by a hierarchical process. The exposition of such stripes to iPrOH results in the formation of helical stripes that finally forms tubular structures.

In addition to the description of such hierarchical supramolecular process, the authors elaborate a chemo-mechanical theoretical model to justify the chirality achieved at the macroscopic level. In general, I consider that this manuscript is suitable to be published in Nature Commun. It is a multidisciplinary work, dealing with a hierarchical, chiral process involving a well-known polymer that has been amply explored by the authors (see, for instance, *Adv. Mater.* 2007, 19, 3353). The results are interesting and timely but some minor points should be addressed prior to its publication:

- 1) The authors utilize equally the concepts of "amplification" and "transfer" of chirality. In my opinion, the authors should be more precise using these two terms. In my opinion, the process investigated in the manuscript deals with a transfer of chirality at different levels: 1) from the CSA to the PANI to generate helical fibers; 2) from the fibers to the stripe and 3) the addition of iPrOH provokes a further helical ribbon. Only in the case of the polymerization of PANI with mixtures of (S)-CSA and (R)-CSA at different enantiomeric excess, the process can be classified as "amplification of chirality". I consider that the authors should prepare a Scheme to clarify the reader the whole process.
- 2) As stated before, the formation of chiral fibers from PANI and CSA is well known and the

authors should incorporate some reference. An example is the above-mentioned Adv. Mater. 2007, 19, 3353 paper. In this regard, the authors should clarify why the chiroptic features described in this manuscript are not the same to that described previously (please, see page 2557 in Chem. Soc. Rev. 2010, 39, 2545.

3) The chemo-mechanical theoretical model should be confirmed with some other example of a chiral acid

4) The effect of iPOH at the molecular level should be clarified. Why the addition of iPrOH provokes the twisting effect on the stripe.

5) Nothing is explained about the addition of chiral 2-aminoethanol. Why is this chiral amine selected? How is the interaction between this chiral amine and the complex PANI-CSA?

In short, I consider this manuscript is suitable for publication in Nature Communications after minor alterations.

Reviewer #3 (Remarks to the Author):

This paper reports interesting observations that millimeter-scale flat stripes, prepared by the self-assembly of a chiral acid (camphor sulfonic acid)-doped polyaniline fibers using a uniaxially stretched polypropylene substrate, undergoes a stepwise transformation into a helically rolled-up structure in response to the change in solvent conditions, where the helical sense is determined by the chirality of the dopant. Although there are numerous examples of the chirality induction of nanoscale molecular assemblies using chiral guests or dopants, this observation clearly demonstrates that the molecular chirality can induce the chirality of a macroscopic material across multiple length scale. Furthermore, the operation of majority rule at the macroscopic scale and chirality-dependent doping and de-doping behavior associated with the motions of the macroscopic stripes are impressive. In addition to these phenomenologically interesting observations, the authors made many sound experiments and theoretical approaches for structural characterization of the material as well as understanding of the mechanism of the helical roll-up, and thus, the conclusions are convincing. I feel these results, which might be related to not only fundamental issue of asymmetry in macroscopic natural systems but also the design of soft actuator, will appeal to the broad readership of the journal. Therefore, I am happy to recommend this paper in the journal. However, I think the authors should address the following issues (mainly the definition of terminology) and revise the manuscript accordingly.

1. Although the authors sometimes use "symmetry breaking" and "chirality amplification" throughout the manuscript, these terms cannot be used to describe the most of present observations, except that the operation of majority rule, which is indeed a kind of chirality amplification phenomenon. The occurrence of the helical structure of the stripes and the enhancement of CD signal (g-factor) is just a result of chirality induction. The use of "symmetry breaking" and "chirality amplification" should be avoided (except for the majority rule section). The title and the introductory part should also be revised accordingly.

2. Title: The phrase "Ordered Macroscopic Assemblies" is obscure.

3. Abstract: The phrase "enantiomer-doped" is misleading. For example, I think "enantiomeric CSA-doped" would be better.

4. To facilitate understanding of readers, it is better to explain the preparation method for the macroscopic stripes using illustration.

5. Overall, the quality of figures (spectral data) is not good. In particular, color and text should be revised.

6. Supplementary Information: As the paper "Y. Yan et al., Hierarchical Crystalline Superstructures of Conducting Polymers with Homohelicity. Chem. Eur. J. 16, 8626-8630, 2010" in the SI is closely related to this work, it should be cited in the main text.

7. Supplementary Figure 1: The assignment of the diffractions should be shown in the figures.

8. Supplementary Figure 2: The unit of q should be [\AA^{-1}].

9. A specific comment on the experiments for majority rule expression: Is the helical morphology of the stripes change depending on the enantiomeric excess of CSA?
10. A specific comment on the mechanism shown in Fig. 3a (top): Is it possible to confirm the geometrical change of the polyaniline assembly upon shrinking of the stripe by XRD?

Response to the comments of reviewers:

We would like to thank the referees for spending time on this manuscript and providing helpful comments which helped us improve the quality of the paper. The manuscript has been carefully revised according to the comments.

Reviewer #1:

General comment 1. The authors prepared a kind of macro helical ribbon based on enantiomer-doped polyaniline and discussed this chirality amplification phenomenon from molecular level to macroscopic level. In addition, they also found the helical self-motion of the ribbons under stimuli of chiral molecules. In general, it is a well-organized paper with a good description. As for an extensively-studied chiral conducting system, I don't think the authors have achieved some breakthroughs here.

General Response 1: We appreciate that the referee recognizes the overall quality of the manuscript. The unique helical self-motion under stimuli of chiral species is also pointed out, and this part is further strengthened in the revision.

Moreover, we would like to further explain the novelty and importance of the manuscript, and why chiral conducting system is selected. In this study, conducting polyaniline doped by enantiomeric camphor sulfonic acid as a dopant is selected since its chirality at the levels of molecular and supramolecular has been well studied. The study on the self-assembly of polyaniline has inspired the investigation of various conducting or semiconducting systems with promising applications. However, the macroscopic helical chirality is not realized in conducting polyaniline nor other analogous systems. In the result, we show that molecular chirality can transfers to macroscopic helical chirality by breaking through the scale barriers. An interesting helical self-motion behavior of the stripe under stimuli of chiral species is demonstrated for the first time. This study provides significance in both experimental and theory toward comprehensive understanding the hierarchical chirality transfer and induction in chiral conjugated molecular systems, and even in other functional natural and artificial materials.

Comment 1.1. The authors have discussed the mechanism for the formation of helical ribbons from the view of energy, and clarified the relationship between some important parameters, including microscopic shrinkage strain, bending curvature, torsional curvature, and out-of-plane deformation energy. However, the above discussion part just explained why flat stripes will twist in poor solvent.

The majority rule experiment and CD data are far from revealing the chirality amplification mechanism. Is it possible to find the functional relationship between the molecular-chirality and macro-helicity? The authors should strengthen this part.

Response 1.1. We appreciate the reviewer for reviewing our manuscript meticulously. In the revision, we supplement relevant experiments and calculations to further clarify the chirality amplification phenomenon in the majority rules experiments as well as chirality induction of the whole process. The “**Molecular origin of the macro helical chirality**” of main text was revised according to the additional results and details were summarized in **Supplementary Note 5**. The outlines are listed as followings:

Firstly, the intention of using majority rules experiments and the conclusions from the corresponding chirality amplification phenomenon are cleared and some relevant references are cooperated in this section.

Secondly, the functional relationship between the molecular chirality and macroscopic helicity is summarized in this section. In general, the molecular chirality of enantiomeric CSA-dopant transfers to the macromolecular, supramolecular and micro-structured chirality of PANI-CSA via the principle of supramolecular self-assembly (*e.g.* *Chem.Soc.Rev.*, **2010**, *39*, 2545–2576; *Adv. Mater.*, **2013**, *25*, 6039–6049), and further transfers to macroscopic helical chirality of macroscopic ribbons via the hierarchical chemo-mechanical mechanism explained here. The molecular chirality dictates the helical direction of the supramolecular and microscopic assemblies, and the helical direction of the microscopic assemblies further dictates the helical direction of the macroscopic assemblies. In conclusion, their function relationship is that the molecular chirality dictates the helicity of the final macroscopic assemblies by controlling the helicity of the structures in intermediate hierarchies.

Comment 1.2. The authors mentioned that the difference between the outside and inside surfaces contributed to the helical conversion. Will the choice of substrates for assembly influences the structure or p/d of the macro ribbon?

Response 1.2. We thank the reviewer’s comment. To address this comment, additional experiment is carried out by using uniaxial stretched polyethylene (PE) as substrate. The change of the substrates by using PE instead of PP does not influence the microscopic structures nor the structural feature of the macro ribbons at different Φ_{iPrOH} (Fig. R1.2). In principle, the selected substrate should not be extensively swelled nor dissolved by solvents, and it should be stretchable and have hydrophobic

surface that is appropriate for attached growth of PANI (*P. Natl. Acad. Sci. USA*, **2010**, *107*, 19673-19678). After hydrophilic treatment by plasma, for example, the PANI will not assemble on the substrate (the margin of substrate in the insertion of Figure R1.2d).

Figure R1.2 | PANI:R-CSA stripe assembled on a uniaxial stretched HDPE (high density polyethylene) template. b, Folded corner of the stipe with flat inside surface and helical fibrous outside surface. a and c, The magnified inside and outside surfaces of the stipe from b; d, e, f, Photographs of the right-handed helical ribbons with different feature size of helicity with $\Phi_{i\text{PROH}}$ fixed at ca. 0%, 20% and 40%, respectively. The insertion of b is the stripe adhere to the uniaxial stretched PE substrate.

Comment 1.3. The helical self-motion is very interesting and inspiring. In my opinion, compared with the structure analysis, they should focus on the chirality-responsive property of these materials. For example, the authors can investigate how the structure of chiral molecules, concentration, and reaction time influence the helical conversion of ribbons from the view of thermodynamics and kinetics, which will motivate the study of biomimetic smart materials.

Response 1.3. We appreciate the suggestion from the reviewer, and we supplemented the studies on how the structure of enantiomeric target species, its concentration and de-doping time influence the helical conversion of the macroribbons from the view of thermodynamics and kinetics. These complement studies carried out according to the suggestion further provide an understanding on why the macrostipe demonstrates unique helical self-motion behaviors under stimuli of enantiomeric

target species with opposite chirality. We summarized these complement studies in “**Helical self-motion of the macrostripe under stimuli of enantiomeric amino hexanes**” of the main text and details in **Supplementary Note 6**. Some highly correlated references are cited to expound the importance of this section. The outlines are listed as followings:

Upon adding of the chiral target species, the PANI stripes would de-dope and turn into blue from green, and its UV-vis-NIR spectra would blue shift accordingly (Supplementary Information Fig. 12a, and reference *J. Am. Chem. Soc.* **2004**, *126*, 2278-2279). Therefore, we recorded the de-doping process of the stripe by UV-vis spectra upon adding a series of enantiomeric amines with different initial concentrations (examples see Supplementary Figs. 14 and 15). We use attenuation of peak intensity at 470 nm of the PANI stripe to demarcate its de-doping degree.

For enantiomeric target species with chiral selectivity (2-amino hexane, 2-aminononane, 1-methyl-3-phenylpropylamine, 1-phenylbutylamine, and 2-methyl-1-phenylpropan-1-amine), their constant k_a of target species with compatible chirality are an order of magnitude higher than that of with incompatible chirality ones (Fig. 4b, and Supplementary Table 2).

Regarding the amino as the pivot of the target species, besides its configuration, the substituent groups attached to its two sides influence its k_a by influencing both its migration in PANI assemblies and its bonding to enantiomeric CSA because of steric hindrance effect. As volume of the substituent increases, k_a decreased. Too low [(*S*)-(+)-2-aminobutane] or too high [(1*S*)-2-(4-methylphenyl)-1-phenylethanamine] steric hindrance from the substituent groups may leading the enantiomeric amines lost their selectivity to the stripe. The deformation behavior of the stripe is sensitive to the concentration of the target species, low initial concentration of amino target species only leads to mild irregular deformation of the stripes.

Comment 1.4. The authors should provide the scale bars for Figure 1 more clearly.

Response 1.4. We thank the comment from the reviewer. Accordingly, we uniformed the format of the scale bars in Figure 1 and provide a more detailed scale instruction in the corresponding legend.

Reviewer #2:

General comment 2. The manuscript by Yang et al. describes the hierarchical self-assembly of PANI stripes prepared upon the addition of chiral CSA dopants. The presence of the chiral dopant induces the formation of chiral structures (demonstrated by the corresponding CD spectra). This phenomenon is well-reported in literature, including some references from the same authors not mentioned in the manuscript. The deposition of such PANI-CSA polymers onto a polypropylene substrate allows the preparation of stripes with dissimilar inside and outside surfaces by a hierarchical process. The exposition of such stripes to iPrOH results in the formation of helical stripes that finally forms tubular structures. In addition to the description of such hierarchical supramolecular process, the authors elaborate a chemo-mechanical theoretical model to justify the chirality achieved at the macroscopic level. In general, I consider that this manuscript is suitable to be published in Nature Commun. It is a multidisciplinary work, dealing with a hierarchical, chiral process involving a well-known polymer that has been amply explored by the authors (see, for instance, Adv. Mater. 2007, 19, 3353). The results are interesting and timely but some minor points should be addressed prior to its publication. In short, I consider this manuscript is suitable for publication in Nature Communications after minor alterations.

General response 2. We appreciate that the reviewer spent precious time on our manuscript and recognizes our present work interesting and timely. The manuscript has been carefully revised according to the suggestion from the reviewer.

Comment 2.1. The authors utilize equally the concepts of “amplification” and “transfer” of chirality. In my opinion, the authors should be more precise using these two terms. In my opinion, the process investigated in the manuscript deals with a transfer of chirality at different levels: 1) from the CSA to the PANI to generate helical fibers; 2) from the fibers to the stripe and 3) the addition of iPrOH provokes a further helical ribbon. Only in the case of the polymerization of PANI with mixtures of (S)-CSA and (R)-CSA at different enantiomeric excess, the process can be classified as “amplification of chirality”. I consider that the authors should prepare a Scheme to clarify the reader the whole process.

Response 2.1. We appreciate the reviewer for pointing out our ambiguities using of these key concepts. We checked the whole text and clarified these key concepts, including symmetry breaking, chirality amplification and chirality induction, chirality transfer, mentioned by the reviewer, accordingly. We feel that the process is indeed complex and may puzzle the reader as mentioned by

the reviewer. Therefore, we designed a scheme to clarify the process including the material preparation and macroscopic helical chirality expression (see **Scheme 1b** in the revision). The “**Molecular origin of the macro helical chirality**” of main text was revised according to the suggestion as well.

Comment 2.2. As stated before, the formation of chiral fibers from PANI and CSA is well known and the authors should incorporate some reference. An example is the above-mentioned *Adv. Mater.* 2007, 19, 3353 paper. In this regard, the authors should clarify why the chiroptic features described in this manuscript are not the same to that described previously (please, see page 2557 in *Chem. Soc. Rev.* 2010, 39, 2545).

Response 2.2. We thank the reviewer’s comment. To clarify the difference of chiroptic features in these cases, we incorporate the two mentioned references (see Reference 38 and 39 in the main text). In order to further illuminate the reason that causes these differences, some original references in which chirality induction mechanism of PANI was investigated such as *Polymer*, 1994, 36, 3113; *Macromolecules*, 1998, 31, 6529; *Synth. Met.*, 1994, 66, 93; *Polymer*, 1995, 18, 3597 (see Reference 32 and 35-37 in the main text) are incorporated as well.

The remarkable difference is that the present PANI:CSA structures exhibiting an inverted sign for the character CD bands at *ca.* 450 nm comparing with its electrochemically synthesis and aqueous-phase synthesis analogous in previous reports. This may result from different conformations between chemically synthesized PANI:CSA in organic solvent herein and chemically or electrochemically synthesized PANI:CSA in aqueous-phase (*Macromolecules*, 1998, 31, 6529; *Adv. Mater.* 2007, 19, 3353). The difference in molecular conformations (compact coil or expanded coil) may result in different intermolecular stacking of PANI, which aggravated the difference in their character CD bands of PANI assemblies.

In this manuscript, whereas, the PANI was synthesis by chemical oxidation of aniline in organic solvent using DDQ as an oxidant in the presence of enantiomeric CSA, then the as-prepared PANI were assembled slowly in organic solvent with high G/P ratio. This PANI solution and its assemblies possesses similar chiroptic features compare with emeraldine base doped by CSA in organic solutions (the chemically prepared film in Fig. R2.2a, c, and d).

We summarized this explanation in the last paragraph of “**Basic characters of the macrostripe**” of the main text and cited the corresponding references. Details are added in the last paragraph of

Supplementary Note 2.

Figure R2.2 | CD spectra of PANI:CSA prepared by different synthesis and assemble conditions. a, CD of chemically and electrochemically prepared PANI:*s*-CSA film (*Macromolecules*, 1998, 31, 6529); b, CD of chemically synthesis and assembled PANI nanofibers (*Adv. Mater.* 2007, 19, 3353); c and d, CD of PANI solutions and self-assembled structures prepared in organic solvents in the present manuscript. The wavelength was selected from 300 to 700 nm from Figure 1e and Supplementary Figure 3.

Comment 2.3. The chemo-mechanical theoretical model should be confirmed with some other example of a chiral acid.

Response 2.3. We thank the reviewer's comment. We tried to prepare similar stripe by using chiral acid dopants such as enantiomeric tartaric acid and pyroglutamic acid. Unfortunately, no well-doped PANI solutions are obtained, which prevent for a further self-assembly process.

Experimentally, protonic acids such as enantiomeric tartaric acid and pyroglutamic acid, can be hardly dissolved nor dope aniline monomer in CHCl_3 solvent. At the very first stage, when the DDQ THF solution was added into the above mentioned CHCl_3 solution, the mixed solution changed into

blue quickly. This proved that neither the aniline monomer nor the corresponding polymer can be well doped by tartaric acid nor pyroglutamic acid in situ using CHCl_3 as a solvent. Then the PANI solution changed into dark brown with dark particles from blue quickly. The un-doped PANI and its poor solubility in the mixed organic solvent hinder the subsequent experiments.

Figure R2.3 | The dark brown PANI solutions using enantiomeric tartaric acid and pyroglutamic acid as dopant respectively.

Comment 2.4. The effect of *i*PrOH at the molecular level should be clarified. Why the addition of *i*PrOH provokes the twisting effect on the stripe.

Response 2.4. We thank the reviewer’s helpful comment. At the molecular level, introducing *i*PrOH into THF induced intermolecular shrinkage in the stripe. The intermolecular shrinkage perpendicular to the π -plane of PANI induces the microscopic shrinkage strain along the long axis of the fiber in the stripe. This alteration according to the comment made the logic of our work clearer. We added this explanation in the “**Chirality induction mechanism based the multi-scale chemo-mechanical model**” in the main text. And additional WAXS experiments are added in **Supplementary Figure 7 and Supplementary Note 3**.

Experimentally, we used wide angle X-ray scattering (WAXS) to measure the change of the macrostripe at molecular level in solution upon its shrinkage. The structure of the samples dispersed in the solvents of THF and *i*PrOH can be detected in quartz capillaries, respectively (**Supplementary Fig. 7**). The *d*-spacing of intermolecular π - π stacking of the stripe immersed in the THF and *i*PrOH

are *ca.* 3.468Å and 3.436Å, respectively, and their difference value is *ca.* 0.032 Å. In the calculation of DFT, the change of intermolecular *d*-spacing of π - π stacking is *ca.* 0.03 Å (**Fig. 3a, Supplementary Note 3**). The distance change of intermolecular *d*-spacing of π - π stacking between theoretical and experimental values are congruent. In this way, the effect of iPrOH at the molecular level are clarified in the way of both experimental and theoretical.

Comment 2.5. Nothing is explained about the addition of chiral 2-aminohexane. Why is this chiral amine selected? How is the interaction between this chiral amine and the complex PANI-CSA?

Response 2.5. We thank the reviewer's comment. According to the suggestion, we added the selection reason (Discrimination of enantiomers, such as amines, diamines, alcohols, amino alcohols, amino acids and the like that closely related to pharmaceutical, chemical, biotechnology and bio-safety, environment, and food industries, is one of core research fields in analytical and catalytic chemistry.) in the "**Helical self-motion of the macrostripe under stimuli of enantiomeric amino hexanes**" of main text. Additional experiments were carried out to further explore the interactions and selectivity essence between chiral amines and the PANI:CSA stripes. Details please see **Supplementary Note 6**, and the key points are outlined as following:

Upon adding of the chiral target species, the PANI stripes would de-dope and turn into blue from green, and its UV-vis-NIR spectra would blue shift accordingly (Supplementary Information Fig. 12a, and reference *J. Am. Chem. Soc.* **2004**, *126*, 2278-2279). Therefore, we recorded the de-doping process of the stripe by UV-vis spectra upon adding a series of enantiomeric amines with different initial concentrations (examples see Supplementary Figs. 14 and 15). We use attenuation of peak intensity at 470 nm of the PANI stripe to demarcate its de-doping degree.

For enantiomeric target species with chiral selectivity (2-aminohexane, 2-aminononane, 1-methyl-3-phenylpropylamine, 1-phenylbutylamine, and 2-methyl-1-phenylpropan-1-amine), their constant k_a of target species with compatible chirality are an order of magnitude higher than that of with incompatible chirality ones (Fig. 4b, and Supplementary Table 2).

Regarding the amino as the pivot of the target species, besides its configuration, the substituent groups attached to its two sides influence its k_a by influencing both its migration in PANI assemblies and its bonding to enantiomeric CSA because of steric hindrance effect. As volume of the substituent increases, k_a decreased. Too low [(*S*)-(+)-2-aminobutane] or too high [(1*S*)-2-(4-methylphenyl)-1-phenylethanamine] steric hindrance from the substituent groups may leading the enantiomeric

amines lost their selectivity to the stripe. The deformation behavior of the stripe is sensitive to the concentration of the target species, low initial concentration of amino target species only leads to mild irregular deformation of the stripes.

Reviewer #3:

General comment 3. This paper reports interesting observations that millimeter-scale flat stripes, prepared by the self-assembly of a chiral acid (camphor sulfonic acid)-doped polyaniline fibers using a uniaxially stretched polypropylene substrate, undergoes a stepwise transformation into a helically rolled-up structure in response to the change in solvent conditions, where the helical sense is determined by the chirality of the dopant. Although there are numerous examples of the chirality induction of nanoscale molecular assemblies using chiral guests or dopants, this observation clearly demonstrates that the molecular chirality can induce the chirality of a macroscopic material across multiple length scale. Furthermore, the operation of majority rule at the macroscopic scale and chirality-dependent doping and de-doping behavior associated with the motions of the macroscopic stripes are impressive. In addition to these phenomenologically interesting observations, the authors made many sound experiments and theoretical approaches for structural characterization of the material as well as understanding of the mechanism of the helical roll-up, and thus, the conclusions are convincing. I feel these results, which might be related to not only fundamental issue of asymmetry in macroscopic natural systems but also the design of soft actuator, will appeal to the broad readership of the journal. Therefore, I am happy to recommend this paper in the journal. However, I think the authors should address the following issues (mainly the definition of terminology) and revise the manuscript accordingly.

General response 3. We appreciate that the reviewer spent precious time on our manuscript and recognizes our present work as valid. And the manuscript was revised according to the suggestion from the reviewer, carefully.

Comment 3.1. Although the authors sometimes use “symmetry breaking” and “chirality amplification” throughout the manuscript, these terms cannot be used to describe the most of present observations, except that the operation of majority rule, which is indeed a kind of chirality amplification phenomenon. The occurrence of the helical structure of the stripes and the enhancement of CD signal (g-factor) is just a result of chirality induction. The use of “symmetry breaking” and “chirality amplification” should be avoided (except for the majority rule section). The title and the introduction part should also be revised accordingly.

Response 3.1. We thank the reviewer for pointing out these misused concepts that help us to clear the subject and further improve the manuscript timely. We checked the whole text, and clarified the key concepts, including symmetry breaking, chirality amplification and chirality induction, chirality transfer, mentioned by the reviewer, accordingly. We use “induce” to replace the “chirality amplification” in the title. The “**Molecular origin of the macro helical chirality**” of main text was revised accordingly as well.

Comment 3.2. Title: The phrase "Ordered Macroscopic Assemblies" is obscure.

Response 3.2. We thank the reviewer’s helpful comment. We’ve replace the “ordered macroscopic assemblies” into “hierarchical self-assemblies”. Accordingly, the title is changed to “**Macroscopic Helical Chirality and Self-motion of Hierarchical Self-assemblies Induced by Enantiomeric Small Molecules**”.

Comment 3.3. Abstract: The phrase “enantiomer-doped” is misleading. For example, I think “enantiomeric CSA-doped” would be better.

Response 3.3. We appreciate the referee for reviewing our manuscript meticulously. According to the suggestion, we’ve revised the “enantiomer-doped” into “enantiomeric CSA-doped” in both the abstract and where else it appeared in the manuscript.

Comment 3.4. To facilitate understanding of readers, it is better to explain the preparation method for the macroscopic stripes using illustration.

Response 3.4. We thank the reviewer’s helpful comment. The process is indeed complex and may puzzle the reader. According to the suggestion from the reviewer, we prepared a scheme to clarify the process including the material preparation and macroscopic helical chirality expression (**Scheme 1b** in the revision).

Comment 3.5. Overall, the quality of figures (spectral data) is not good. In particularly, color and text should be revised.

Response 3.5. We thank the reviewer’s helpful comment. According to the suggestion, we revise the text into a precise and simple form in the figures and adjusted the color of all the text into black. We turned up the color contrast of the curves in the spectra of Figure 1, 2 and 3. We hope the gradient change in the color of the lines with ee% or Φ_{IPrOH} can help the readers to figure out the relationships

between the chirality and ee%, or between the chirality and Φ_{IPrOH} , intuitively.

Comment 3.6. Supplementary Information: As the paper "Y. Yan et al., Hierarchical Crystalline Superstructures of Conducting Polymers with Homohelicity. *Chem. Eur. J.* 16, 8626-8630, 2010" in the SI is closely related to this work, it should be cited in the main text.

Response 3.6. We thank the reviewer's helpful comment. According to the suggestion, we've moved the citation (Y. Yan et al., Hierarchical Crystalline Superstructures of Conducting Polymers with Homohelicity. *Chem. Eur. J.* 16, 8626-8630, **2010**) to the main text of the revision (please see reference 34).

Comment 3.7. Supplementary Figure 1: The assignment of the diffractions should be shown in the figures.

Response 3.7. Thanks for the referee's kind suggestion, and according to the suggestion, we have assigned the diffractions in the figures and renewed the Supplementary Figure 1.

Comment 3.8. Supplementary Figure 2: The unit of q should be [\AA^{-1}].

Response 3.8. Thanks for the referee's kind suggestion, and we corrected this mistake in the revision.

Comment 3.9. A specific comment on the experiments for majority rule expression: Is the helical morphology of the stripes change depending on the enantiomeric excess of CSA?

Response 3.9. We thank the reviewer for the comment. The helical morphology of the stripes changed depending on the enantiomeric excess of CSA. In detail, the enantiomeric excess of CSA determines the proportion of left- and right-handed helical locus in the microfibers, nonlinearly. We recorded the SEM images of the stripes doped by CSA with various enantiomeric excess of *s*-CSA (ee% of *s*-CSA), additionally, and summarized the relation between statistic proportion of left-handed and right-handed helical locus and the enantiomeric excess of CSA (**Supplementary Fig. 10**). The net helicity of the microscopic fibers to the enantiomeric excess of CSA satisfies likewise the majority rules (**Supplementary Fig. 11a**). The reason for this phenomenon, please see the additional explanation about the majority rules experiments in **Supplementary Note 5**.

Comment 3.10. A specific comment on the mechanism shown in Fig. 3a (top): Is it possible to confirm the geometrical change of the polyaniline assembly upon shrinking of the stripe by XRD?

Response 3.10. We thank the reviewer for the comment. We tried to use wide angle X-ray scattering (WAXS) measuring the change of the macrostripe in solution upon its shrinkage, additionally. In this way, samples dispersed in the solvents of THF and iPrOH can be measured in quartz capillaries, respectively (**Supplementary Fig. 7**). The *d*-spacing of intermolecular π - π stacking of the stripe immersed in the THF and iPrOH are *ca.* 3.468Å and 3.436Å, respectively, and their difference value is *ca.* 0.032 Å. The distance change of intermolecular *d*-spacing of π - π stacking between theoretical and experimental values are congruent. We added this experimental results and corresponding explanation in the **Supplementary Note 3**.

REVIEWERS' COMMENTS:

Reviewer #1 (Remarks to the Author):

I think the authors have well explained the molecular origin of the macro helical chirality with additional experiments and addressed the reviewers' comments. I would like to would like to recommend publishing this work.

Reviewer #2 (Remarks to the Author):

I consider that the authors have made considerable efforts to justify and clarify all the remarks raised by the referees and, especially, for all the remarks I did. Therefore, I consider that the manuscript meets the quality and novelty criteria to deserve to be published in Nature Communications.

Reviewer #3 (Remarks to the Author):

I found that the authors have properly addressed my comments and revised the manuscript accordingly. Now I recommend publication of this paper in the journal without any concern.